# *Schēma*: A Semantic Puzzle—Some Hermeneutical and Translational Difficulties about Philippians 2:7d

Teresa Bartolomei 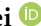

Research Centre for Theology and Religious Studies (CITER), Catholic University of Portugal (UCP), Campus Palma de Cima, P-1649-023 Lisbon, Portugal; tbartolomei@ucp.pt

**Abstract:** The occurrence of the term σχήμα in Phil 2:7d is analyzed in comparison with two other crucial Pauline occurrences: 1 Cor 7:31 and Phil 3:21 (here as a semanteme included in the verb μετασχηματίσει). This comparative study aims to provide a revision of the current interpretation of the word as designating the outward, sensory, accidental appearance in which Christ's human nature was manifested to those who dealt with him. This traditional reconstruction is unsatisfactory in two respects: (1) it is tributary to a substantialist ontology that identifies corporeality as a mere spatial extension, unrelated to historicity and (2) it is fraught with highly problematic theological, potentially docetic, implications. As an alternative, the term σχήμα is here interpreted within the framework of the great Pauline theology of history: as a temporal–eschatological marker designating the peculiar temporal state of transience and suffering corruptibility inherent in physicality and corporeal life. This change also clarifies the conceptual articulation of σχήμα with the parallel expression μορφὴν δούλου. According to this interpretation, contrary to the prevailing view, the locution "slave form" does not designate 'the' or 'one' 'human form' but the 'creature form', as cosmic submission to temporal finitude.

**Keywords:** Philippians (epistle); Paul; σχήμα; μορφὴν δούλου; time

## 1. Introduction

The pericope commonly referred to as the Christological hymn[1] of Philippians 2:5-11, a text of great exegetical and theological importance, is at the center of an intense interpretative debate and of ever new translation proposals. The main challenge is the handling of a field of lexical indeterminacy carved out by the use of a particularly laborious terminology, in which Judaic–Old Testamentary and classical, mainly Hellenistic, semantic codes intersect. An auroral and magisterial station in that process of the inculturation of the proclamation of the historical Jesus, of which Paul is the protagonist and first apostle (Penna 2002, p. 57),[2] the pericope reveals a dizzying genealogical–etymological complexity, entrusted to the formulation of a message of revolutionary novelty and originality, a bearer of some of the founding elements of a Trinitarian Christology (cf., Martin and Dodd 1998). The tension between the stratigraphic acuity of the lexical analysis and the broad theological perspective required by this handful of verses makes its reading and translation particularly difficult, inviting the interpreter to the very exercise of humility that is the main object of the appeal to the Christians of Philippi in which the pericope is embedded. No overall interpretation, no translation of Philippians 2:5-11[3] has yet arrived at an undisputed, universally accepted, and acceptable solution and version. It is therefore "regarding others as more **significant** than ourselves" (2:3)[4] that this paper focuses on a specific lexical question, in the hope that such an analysis, while not providing new answers, will offer a useful critical key to further our understanding of this essential text.

## 2. An Elusive Word: σχήμα

The main objective of this reflection is indeed to reconstruct the peculiar semantic value that the Greek term σχήμα (*schēma*) (2, 7d)[5] assumes in the Pauline corpus, to show

that only through this comparative study can we determine its correct interpretation and translation in the body of the pericope.

Two converging options in the extensive exegetical literature on Phil 2:5-11 have so far undermined an adequate understanding of this word. First of all, there has been a clear conceptual subordination of it in relation to the term μορφὴ (*morphē): the meaning of *schēma* is thus often 'deduced' from the interpretation of the former (in its main semantic core of "appearance, visible form"), sometimes even being reduced to a second-degree, weak synonym, in a misleading reading of the text.[6] Second, it is not taken into account analytically (it is often barely mentioned) the Pauline occurrence of the word in 1 Cor 7:31, in a collocation of very strong semantic poignancy, commensurate with its conceptual significance in Philippians 2:7d, and consistent with the use of the semanteme (incorporated in the verb *metaschēmatizei*) in Phil 3:21 and Rom 12:2 (*syschēmatizei*).

An eloquent symptom of this exegetical chiaroscuro is the translational uncertainty associated with the locution σχήματι εὑρεθεὶς ὡς ἄνθρωπος, attested by a discrepancy and variability of solutions that make this and the (much more studied but still not fully resolved) expression ἐν ὁμοιώματι ἀνθρώπων (2, 7c) a real linguistic *crux*.

### 3. The Interpretation of σχήμα as "Outward Appearance"

*Schēma*, which the Latin of the Vulgate translates respectively with the term *habitus* in Phil 2:7d and with *figura* in 1 Cor 7:31, is a rare and 'cultured' term,[7] gravitating in the philosophical and rhetorical area,[8] whose import into New Testament Greek therefore has a 'technical' density that cannot be underestimated. The fact that Paul uses it in parallel as an identifying qualifier of the world and of the human being ("schēma of the cosmos" in Corinthians, "schēma of man" in Philippians), a general designation of a (physically marked) 'phenomenological form' common to both, seems intriguing and surprising. It suggests that such a pairing is indicative of a precise semantic connotation which is unexpressed and even concealed in current translations. With a philological background strongly conditioned by the substantivist–ontological categories of classical metaphysics,[9] they read the term simply as *external appearance*;[10] *visible aspect*;[11] a sensory, material *mode of manifestation*[12] (TDNT 1971, p. 956), possibly in contrast to an abstract, conceptual notion of form.

The insurmountable difficulty associated with the exegetically unsatisfactory determination of *schēma* as the *outward appearance* of the *human nature-form* of Christ (of the 'natural', 'earthly' *nature-form* of the world?) is that it conceals and implies an insidious separation between the human, substantial nature of Jesus, and his accidental, "empirical manifestation", as the vector of Jesus' identification as a human being by other human beings. This split reappears insistently in translations, lexicons, and commentaries and is clearly the fruit of an ontological pre-comprehension, of a substantivist stamp,[13] not free from a residual dualistic bias, and exposed to undesirable docetic drifts.[14] Based on this lexical interpretation, the original text does indeed seem to circumscribe an intersubjective space of the identification of Christ's humanity as *outward appearance*, which runs the risk of reading the "similarity" (*homoíōma*) invoked in the previous locution in a misleading way, that is as a parallel, an analogy, and not as the denotation of an essential sharing of human nature. If the text does say that Jesus was "identified as man in his outward appearance", then it seems to suggest that this coming into the world "in the likeness of men" means a similarity and not an identity.[15]

The risk is not merely hypothetical. There are plenty of examples of translation disasters caused by this semantic interpretation, such as the unfortunate version adopted by CEI (2008) (and accepted by BG, the Italian edition of the *Jerusalem Bible*): "diventando simile agli uomini. Dall'aspetto riconosciuto come uomo" ("becoming like men. From the aspect recognized as man", *m.tr.*), in which it is literally written that the Son of God did not become man but became "like men", having (at least) their "aspect". *Does he look like us* or *is he one of us*?[16] To say that someone *has the aspect of a man* (that he looks like a man) does not imply that he might not be a man? The insidiousness of camouflage, which the

semanteme also connotes in Pauline usage[17] (the pagan idea of a God who takes on human appearance, like the ancient gods who visited the earth), is strongly evoked here, barely countered by exegetical distinctions that are not entirely effective.

See, for example, how TDNT entangles itself in this difficulty, by referring, in a *petitio principii*, to various exegetical commentaries (recalled in the footnotes: from Dibelius to Käsemann and Lohmeyer) for which it is supposed to provide the philological basis. It explains in a convoluted form that the "outward appearance" is not to be seen as separate from the rational, inner essence or, worse, as a disguise of it: "The reference is to His whole nature and manner as man. In this respect, the outward "bearing", which He assumes corresponds to His inner being" (TDNT 1971, p. 956). If we are thus reassured that the meaning of the text is that Jesus was a real man and not just apparently so (he was not merely "taken" to be a man because of his outward appearance), it remains to explain why Paul[18] would have used the awkward circumlocution σχήματι εὑρεθεὶς ὡς ἄνθρωπος to assert this, rather than a more straightforward formulation. The argument spirals into a tautological circularity of frustrating inconclusiveness: "There is special stress on the fact that throughout His life, even to the death on the cross, Jesus *was in the humanity demonstrated by His earthly form*. The εὑρεθεὶς expresses the truth that this fact could be seen by anybody, σχήμα does not merely indicate the coming of Jesus, or His physical constitution, or the natural determination of His earthly life, or the shape of His moral character. It denotes the «*mode of manifestation*»." (ibid., p. 956, *m.e.*). What does it mean to say that Christ's humanity was "*demonstrated* by His earthly form"? Should it not rather be said that Christ's humanity "consists" in having the earthly form of a man? In what sense then does man's being not coincide with "the mode of his manifestation", which is designated by the word σχήμα?

**4. The Occurrence of σχήμα in 1 Corinthians 7:31**

In the light of these considerations, and of the relevant and theologically sensitive problems associated with the translation of *schēma* as "outward appearance, visible aspect, mode of manifestation", we must ask ourselves whether this lexical interpretation is not vitiated by a fundamental misunderstanding and whether it is not necessary to radically reconsider the semantic value of the Greek term in Pauline usage. A comparative analysis is therefore recommended, starting with a comparison with the parallel verse 1 Cor 7:31, which concludes the famous sequence of the ὡς μὴ: *as not*, opened by verse 7:29 on the eschatological coming to an end of time:

29 Τοῦτο δέ φημι, ἀδελφοί, ὁ καιρὸς συνεσταλμένος ἐστίν· °τὸ λοιπὸν ἵνα καὶ

Hoc itaque dico, fratres: tempus breve est: reliquum est, ut

I tell you, brothers, the time is running out. From now on

31 παράγει γὰρ τὸ σχῆμα τοῦ κόσμου τούτου.

præterit enim figura hujus mundi.

**For the present fashion of world is passing away**.[19]

When Augustine, commenting on this text, emphasizes that for the Apostle, "it is not the nature of the world that is passing away but its figure" ("Figura enim praeterit, non natura"),[20] the reader is perplexed to ask what is the nature of the cosmos that does not pass away in the fulfilment of the Parousia, in the passing (in the coming to an end) of time (of the καιρὸς). Can one speak of the *physis* of the cosmos as such (of its "imperishable essence") freed from its, already ceasing, "natural appearance"? Can one speak so, given the biblical perspective, which presents creation, unlike the Greek cosmos, as an evolutionary process with a historical profile, rather than as an unchanging mythical/physical entity? In the Bible, the cosmos is a manifestation of the dynamics of a profound transformation of the creature through the intervention of both the Creator[21] and the creature itself. As a matter of fact, human choices also have a radical effect on the state of creation: fallen in the fall of the "earthly man" Adam, redeemed in the gift of himself by the "heavenly man" Christ

Jesus (1 Cor 15: 45–49), who inaugurates and makes possible the coming of the Kingdom of God and the messianic recapitulation of all things in the Son of God made man.

No, the "*ceasing figure of* the cosmos" cannot be biblically reduced to a mere "external manifestation", a visible appearance, an accidental form of an unchanging essence, as if the coming of the Parousia implied nothing more than a restyling, a superficial change that does not affect the profound reality of beings, of creatures. The term $\sigma\chi\acute{\eta}\mu\alpha$ must therefore have a denser meaning than the commonly intercepted one of "external appearance", a meaning that is highlighted by the opening verse of the eschatological discourse on the *as not*. If "the time is running out" ($\acute{o}$ $\kappa\alpha\iota\rho\grave{o}\varsigma$ $\sigma\upsilon\nu\epsilon\sigma\tau\alpha\lambda\mu\acute{\epsilon}\nu\sigma\varsigma$), is coming to an end (whereby the countdown of its passing away has begun, by reckoning what is left of it: $\tau\grave{o}$ $\lambda\sigma\iota\pi\grave{o}\nu$), that which "is passing away" ($\pi\alpha\rho\acute{\alpha}\gamma\epsilon\iota$) is nothing other than the passing away itself, time itself as passing away ($\sigma\chi\acute{\eta}\mu\alpha$). What is "*fading away*" is not the visible, "sensory aspect" of the world but the temporal condition of creation as a "passing away, disappearing", as a condition of being exposed to the end which affects all finite beings.

## 5. The Creaturely Condition and Its Redemption

We have some intertextual evidence that the Pauline use of the term and semanteme $\sigma\chi\acute{\eta}\mu\alpha$ innovatively privileges (in a kind of semantic neologism) the denotation of the dimension of becoming, of the passage and change of time, with its corrupting charge, inherent in the finitude of creatures, their contingency, and their exposure to loss and diminution (as the vector of evil). Although not absent from the semantic horizon of the letters (cf. 2 Cor 11:13–15), the connotation (generally prevalent in non-Pauline occurrences of the term) of the *external*, material, accidental *appearance* as opposed to the substantial, rational, and therefore unchanging nature (*ousía*, as manifested in the *morphē*) of entities is therefore secondary.

A crucial confirmation comes from the comparison with the Pauline description of the cosmic condition of creatures as radical contingency: the condition of that which is changing, passing, disappearing,[22] subject to the law of corruption, which is together the law of death and sin.[23] No creature is necessary; nothing that is and happens in the cosmos is necessary: everything can end; everything does nothing but end, become different, become less, decays in the corrosive power of evil, which is not a positive entity but a condition of subtraction and decline. Precisely this radical contingency is the sign of the creaturely finitude, which qualifies the cosmos and the human being in the sign of corruption, of mortality. Aristotle says so,[24] expressing a classical, pagan, vision; the Bible says so (in the sequence of the sapiential texts, from Job to the Psalms to Ecclesiastes). Paul says so, when, in Romans 8:18–23, a cosmic fresco of extraordinary power, he describes creation as "suffering" the end and corruption, as being subject to becoming as a fatal engine of decay. Becoming, as the source of decline and destruction, 'reigns', even to the point of subjecting creatures to the supreme condition of slavery, that of death ("wages of sin", Rom 6:23), in which the subject loses the availability of the indispensable condition for being considered as a subject: its status as a living being.

> I consider that the sufferings of this present time are as nothing compared with the glory to be revealed for us. For creation awaits with eager expectation the revelation of the children of God; for creation was made *subject to **vanity**,*[25] not of its own accord but because of the one who subjected it, in hope *that creation itself would be set free from slavery to corruption and share in the glorious freedom of the children of God*. We know that all creation is groaning in labour pains even until now; and not only that, but we ourselves, who have the first fruits of the Spirit, we also groan within ourselves as we wait for adoption, the redemption of our bodies (*m.e.*)

Even if not linearly interpretable, the correspondence of this grandiose eschatological picture with the scenario presented in 1 Cor 7:31, Phil 2:5-11, and 3:21 is undeniable. In fact, the authentic meaning of these three Pauline pages can only be grasped by reading each text in the light of the other two. Romans 8:18–23 makes explicit the temporal sense of the

passing away, of the passing away of *this world*, as the passing away of transience, as the stigma of becoming, the crucial dimension of the enslavement not only of human beings but of the whole cosmos to the power of evil, to the morally and physically destructive dynamic of corruption.[26] The whole creation (πᾶσα ἡ κτίσις) "was made subject to vanity" (τῇ γὰρ ματαιότητι ἡ κτίσις ὑπετάγη), to the dynamics of consumption (of corruption) inherent in temporal becoming, and hopes to be liberated from this slavery *(ἐλευθερωθήσεται ἀπὸ τῆς δουλείας τῆς φθορᾶς: ipsa creatura liberabitur a servitute corruptionis in libertatem gloriae filiorum Dei*), which constitutes the *figura*/the *schēma*/the actual phenomenological form of the world. The universal creaturely hope is, therefore, that evil and endings, as subservience to the corrupting power of finitude, will disappear, that the passage of time and the creaturely powerlessness it implies will pass away, thanks to that *redemption of bodies*[27] which is brought about as adoption as children of God. In this eschatological perspective, hope, not only human but cosmic, is the overcoming of one's *schēmatic* condition of creaturely subservience to decay. It is hope in a "victory" of life over death, which is produced as the universal and definitive establishment and recognition of the sovereignty of Christ, the Risen One, over death:

> Because of this, God greatly exalted him
>> and bestowed on him the name
>> that is above every name,
>> that at the name of Jesus
> every knee should bend,
>> of those in heaven and on earth and under the earth,
>> and every tongue confess that
>> Jesus Christ is Lord,
>> to the glory of God the Father.

(Phil 2:9-11)

> [The Lord Jesus Christ] will **refigure** our **body of humiliation** to conform with his glorified body by the power that enables him also to bring all things into subjection to himself.

> ὃς μετασχηματίσει τὸ σῶμα τῆς ταπεινώσεως ἡμῶν σύμμορφον τῷ σώματι τῆς δόξης αὐτοῦ, κατὰ τὴν ἐνέργειαν τοῦ δύνασθαι αὐτὸν καὶ ὑποτάξαι ⸀αὐτῷ τὰ πάντα.

(Phil 3:21)

When the *schēmatic* time (the fallen time of subjugation to sin and death) will have passed away because the fullness of time, the Parousia, has been reached, then "all things, all creation, every knee, every tongue", will submit to that God who by becoming a creature (by incarnating as a man, dying, and rising again), has overcome sin and death and set creatures free from them.

It is indeed evident that what Romans 8:21 as Alain de Lille in his famous Carmen Omnis mundi creatura. calls "corruption" (decay: φθορᾶς, *corruptionis*), as a synonym for "*vanity*" (*vanitas*: ματαιότητι), is the "death" of Phil 2:8, as the epitome of evil, as the culmination of that emptying humiliation, which is being the world and being in the world with the "status of a slave" (*morphē doulou* of Phil 2:7b),[28] in a state of submission to loss and corruption. Redemption for Paul is liberation, in full fidelity to the vision of the biblical God as the liberator, and, just as biblically, it cannot be understood metaphysically but only historically.

On the Judeo-Christian horizon, evil is not a metaphysical entity but a historical–eschatological process, inscribed in *schēmatic*, pre-messianic temporality. The creaturely form of slavery (μορφὴν δούλου) cannot be thought of as a form in the Platonic–Aristotelian sense, as a rational architecture of being, but as a *pre-historic* universal state, mysteriously emerging between a present of destitution and a future of liberation (*vanitati enim creatura*

*subiecta est, non volens sed propter eum, qui subiecit, in spem, quia et ipsa creatura liberabitur*) and coming to fulfilment as the "redemption of our bodies". Liberation is not the Platonic emancipation of the immortal soul from the mortal body, of the idea from matter, of the rational form from the empirical appearance. It is not the exit of the individual from the physical world and its cosmic end ("it is not the world that is passing away but a figure of it", a temporal dimension of it, a form of happening). Redemption, as liberation, is realized as *bodily metaschēmatization*. It is the transformation of the condition of the *body-schēma*, a "carnal", animal body, humiliated by creaturely bondage (τῆς ταπεινώσεως) into a sovereign, spiritual, glorious body (τῆς δόξης), reshaped (σύμμορφον) by a condition of power (ἐνέργειαν) that frees him from submission, to make him filially participant in the cosmic sovereignty of the Son of God (τοῦ δύνασθαι αὐτὸν καὶ ὑποτάξαι ⌜αὐτῷ τὰ πάντα):

> So also is the resurrection of the dead. It is sown corruptible *(ἐν φθορᾷ)*; it is raised incorruptible *(ἐν ἀφθαρσίᾳ)*. It is sown dishonorable *(ἐν ἀτιμίᾳ)*; it is raised glorious *(ἐν δόξῃ)*. It is sown weak *(ἐν ἀσθενείᾳ)*; it is raised powerful *(ἐν δυνάμει)*. It is sown a natural body (*corpus animale: σῶμα ψυχικόν*); it is raised a spiritual body (*corpus spiritale: σῶμα πνευματικόν*). If there is a natural body, there is also a spiritual one. So, too, it is written, 'The first man, Adam, became *(ἐγένετο: factus est)* a living being *(εἰς ψυχὴν ζῶσαν)*,'; the last Adam a life-giving spirit *(εἰς πνεῦμα ζῳοποιοῦν)*. But the spiritual was not first; rather the natural and then the spiritual. The first man was from the earth, earthly; the second man, from heaven. As was the earthly one, so also are the earthly, and as is the heavenly one, so also are the heavenly. Just as we have borne the image of the earthly one *(εἰκόνα τοῦ χοϊκοῦ)*, we shall also bear the image of the heavenly one. *(τὴν εἰκόνα τοῦ ἐπουρανίου)*.
>
> (1 Cor 15:42–49)

If the *schēma hōs anthrōpos* is creaturely part of the *schēma tou kosmou*, then the messianic transformation of the *schēma hōs anthrōpos* brings about a transformation of the *schēma tou kosmou* itself, namely, precisely its passing, its fading away as *schēma*. The whole of creation awaits "the redemption of our bodies" (τὴν ἀπολύτρωσιν τοῦ σώματος ἡμῶν), which makes them "conform" (*summorphon*) with that of the Risen One and operates as a transfiguration (*metaschēmatizei*) of the "natural body" (*schēmatic, ψυχικόν*), "sown corruptible, dishonourable and weak") into a "spiritual body" (*πνευματικόν*), which "raises incorruptible", free from the destructive power of transience.

It is therefore in the light of this combined reading of its occurrences in the Pauline texts that we can grasp the authentic meaning of the word *schēma* in the Philippian pericope. It does not denote a visible aspect, an external appearance, but an intrinsic '*physical condition*' of being part of the world, of being in the world (of "being born") as a human being. *Schēma* is not the body (*soma*), nor its mere outward appearance, but the pre-messianic temporal condition of physicality (of man and the cosmos), to which Jesus subjected himself and which he eschatologically redeemed through his resurrection. Corporeality cannot be reduced to pure spatiality (*res extensa*) but must be recognized phenomenologically as temporality, a historical processuality capable of eschatological self-transcendence in the form of the new (glorious) temporality instituted by the Parousia. At the Second Coming of Christ as the expected Savior (3, 20), the process of redemption already initiated by his Resurrection will be finally and perfectly completed: the *schēmatic*, "caducous, enslaved" spatio-temporal present form of the world and of the human being, which is already receding, will then definitively overrun.[29]

### 6. Creatural Form, Human Condition

| | |
|---|---|
| 5τοῦτο ⌜φρονεῖτε ἐν ὑμῖν ὃ καὶ ἐν Χριστῷ Ἰησοῦ, | 5 Have among yourselves the same attitude that is also yours in Christ Jesus, |
| 6 ὃς ἐν μορφῇ θεοῦ ὑπάρχων | 6 Who, while subsisting in the form of God, |
| οὐχ ἁρπαγμὸν ἡγήσατο | did not regard equality with God |
| τὸ εἶναι ἴσα θεῷ, | **a booty to keep for himself.** |
| 7ἀλλὰ ἑαυτὸν ἐκένωσεν | 7a Rather, he emptied himself, |
| μορφὴν δούλου λαβών, | 7b taking the form of a slave |
| ἐν ὁμοιώματι ἀνθρώπων γενόμενος· | 7c **and in being born, in human likeness**, |
| καὶ σχήματι εὑρεθεὶς ὡς ἄνθρωπος | 7d **found in the human condition,** |
| 8ἐταπείνωσεν ἑαυτὸν | 8a he humbled himself, |
| γενόμενος ὑπήκοος μέχρι θανάτου, | 8b becoming obedient to death, |
| θανάτου δὲ σταυροῦ· | 8c even death on a cross.[30] |
| (NA 28, p. 606) | |

Jesus, has always been (*hyparchōn*)[31] God: he has always had a divine status. *Morphē* means here the form as the primary manifestation of being, its intrinsic "order", a general status that qualifies its manifestation.[32] But when he was *born* as a man, when he was "made" man ("in similitudinem hominum *factus*", as the Vulgate translates), he assumed the form, the condition, of a slave (the one who is submissive, who must "obey" those who are stronger than him). He assumed the form, the status, of a creature.

*Morphē doulou* therefore does not yet designate the incarnation into man.[33] This point is the subject of the utterance of 7c and 7d, as the emphatic reiteration of the word *anthrōpōs* makes clear, and it is conveyed by the binomial *homoiōmati* and *schēmati*. *Morphē doulou* denotes the general precondition of the incarnation as a human being, which is that of the spoliation of the divine statute in its eternity, and the acceptance of the statute, the creaturely form, of flesh (*sarx*) enslaved to the pre-messianic temporal law of caducity and the corruption of time, which is the *schēmatic* condition, the formal architecture of the cosmos in its contingency and finitude.

Kenotic emptying is "not to keep eternity as the supreme freedom (the intangibility of one's own life) for oneself" as a precious and exclusive "booty". It is to renounce it in order to enter the cosmos, the totality of creation that does not "include" its Creator,[34] in its temporal dimension of "suffering", vulnerability, loss, powerlessness, and decay. The eternal, the "ante saecula genitus", enters "in the days of the flesh",[35] as the Letter to the Hebrews says. He learns the obedience proper to the condition of a slave, stripping himself of his condition as Son, submitting to all that this *skhēmatic* and earthly dimension of temporality entails: "In the days when he was in the flesh, he offered prayers and supplications with loud cries and tears to the one who was able to save him from death, and he was heard because of his reverence. Son though he was, he learned obedience from what he suffered" (Heb 5:7–8).

Thus, the *ante saecula genitus* is *genitus* ("is made" in a birth) "in human likeness", to find himself subdued in the earthly condition of a temporal carnal body (*skhēma*), "body of death" (σώματος τοῦ θανάτου) (Rom 7:24), "body of humiliation" (3:21), body of submission to that temporal *skhēmatic* condition which is proper to the *world* (*cosmos*), which is its *figure*: its phenomenological, physical, and historical form. This is why the text does not simply say, as we would expect, that Jesus was born in a human body (*soma*) but specifies that in being born, he came to find himself and therefore "to be found" (*heuretheis*)[36] in a *skhēmatic body*. This is a pre-messianic body, the form of the body that is the object of our natural, earthly experience (the form in which we know each other). It is a transitory form, destined to be transfigured (*metaskhēmatizei*: 'to come out of the *skhēma*'), shedding the humiliation of servitude to assume the glorious form of the sons of God (3:21).

It is precisely for this reason that the text literally emphasizes the parallel between the incarnation of Jesus and the creation of man, choosing the key term of *likeness* to describe the mode of generation, to highlight that salvation is a second creation. If, of all creatures,

the human being was "made in God's image, after God's likeness" (Gen 1:26),[37] the God who incarnates makes himself a *creature* in the "likeness of men", choosing to "be made" like them,[38] to "be born as a human being".[39] For Paul, the incarnation of Jesus constitutes the generation of the new Adam: whose creation is a *figure* (*typos*) of the incarnation of Jesus, the Messiah (the *new* and *last* Adam), in the parallel gesture in which God "makes the human beings" (after his likeness) and "becomes man" ("makes himself after human likeness"). By accepting to be born as men are born, Jesus accepts to die as men die, submitting himself to the cosmic law of time and sin. In this way, however, the *schēmatic* body in which He became man receives from him the life-giving (victorious over death) and sanctifying (victorious over sin) power of the Eternal. The risen Christ transfigures (*metaschēmatizes*) the flesh, the "form of the servant", the "body of humiliation", into the likeness (*summorphon*) of the form of God, the form of the Eternal (of glory). Through his incarnation and resurrection, Jesus Christ frees creatures from bondage and works a second creation: the first man, the *earthly* man (*animal body*: *skhēmatic*), is buried (dies) and is reborn as the *second man*, the *heavenly* man (*spiritual* body).

## 7. Conclusions

The conditions of the life of the human being, of all the human beings (their *figure*, the '*schēmatic*' *body* of their "days of flesh"), of all creatures,[40] are transformed (*metaschēmatized*) in the Risen One through their salvific incorporation into the conditions of the life of the Son of God (who abolishes all servitude, Jn 15:15).[41] In the union with the Risen One, the *figure of man* (the *schēma* in which Jesus *found himself in the world as a human being*) *is passing away* exactly like the *figure of this world* (the shortening of time), in a common eschatological "transformation" (*metaschēmatization*). The Risen One is, actually, already the *Parousia*: the messianic overcoming of the *figure of this world* through the establishment of the Kingdom of God, the "world that is to come", according to the traditional rabbinical distinction, accepted by the early Church.[42] This 'new world', the Kingdom of Heaven, will last "in the ages of the ages" (εἰς τοὺς αἰῶνας τῶν αἰώνων): a 'forever' that does not mean an indefinite duration as an unlimited protraction of time passage but the incorruptible actuality of the present ("subtracted from the *schēmatic* subtraction" of its actuality). The Kingdom of God is not a reign of pure spirits: at the Parousia, the *figure of this world, the schēmatic form of creaturely temporality*, ceases, not because corporeality fades but because its temporal form is transformed, gloriously enlivened by communion with divine eternity. The presence of the Lord is given as a presence that coincides with itself, that establishes the fullness of time as the fullness of a present, whose actuality is not *schēmatically* articulated in relation to the past and the future but is definitively fulfilled in itself.

The *schēma*, the bodily form in which Jesus became man through incarnation, is not the mere "outward appearance", empirical, of a substantial and unchanging nature, but the changeable temporal condition that defines the pre-messianic existence and therefore the essence, of the human being and of the cosmos (τῷ αἰῶνι τούτῳ: *huic seculo*) before Redemption starts its saving work of a second creation. Creaturely enslaved to evil and death, to the physical and moral corruption, *schēmatic* corporeality is the transitional human and cosmic condition, awaiting the redemptive coming of the Kingdom of God, of the Parousia: the glorious communion of all creatures with the Eternal.

**Funding:** This research received no external funding.

**Institutional Review Board Statement:** Not applicable.

**Informed Consent Statement:** Not applicable.

**Data Availability Statement:** No new data were created or analyzed in this study. Data sharing is not applicable to this article.

**Conflicts of Interest:** The author declares no conflicts of interest.

## Notes

[1] It is a matter of controversy among exegetes whether the combination of the seven verses in question, often graphically transcribed as a poetic unit (beginning with Lohmeyer's proposal in 1961), is really a hymn, possibly of liturgical origin and incorporated by Paul into his own letter but not entirely by his own hand, or whether it is rather an encomium or a true confession of faith with a catechetical or liturgical profile. For all the exegetical information and discussions that are recorded without specific reference to sources in this paper, which is selectively devoted to the in-depth study of a particular lexical issue, I refer to the exhaustive critical–bibliographical summaries elaborated in (O'Brien 1991; Martin 1997; Reumann 2008; Fewster 2015; Bird and Gupta 2020). For a glimpse of new exegetical and historical–critical perspectives marked by feminist and postcolonial keys, see (Marchal [2014] 2017).

[2] The Greek of the pericope has such major lexical and syntactical anomalies as to suggest the hypothesis (albeit a minority one) that the text handed down in the Pauline epistle is a translation of a pre-existing Aramaic original (cf., Fitzmeyer 1988).

[3] For the sake of fluidity, the acronym Philippians is omitted in subsequent mentions of individual verses from the second chapter of the epistle.

[4] All Scripture quotations (except for alterations marked in bold) are from NABRE; the Latin version occasionally quoted is from VC.

[5] The division of the pericope into verses is a matter of some controversy among exegetes, who provide different reconstructions of its metrical form. In some editions, the locution καὶ σχήματι εὑρεθεὶς ὡς ἄνθρωπος is included in verse 2, 8, in others in 2, 7. Here, I adopt the criterion of NA28 (from which all quotations in Greek are taken), citing it as 2,7d.

[6] Cf. the choice of the translation of Bird and Gupta (2020, p. 71), who (similarly to the ESV) literally assimilate μορφὴ and σχήμα, translating both with the term "form": "who, though he was in the form of God, did not regard equality with God as something to be exploited, but emptied himself, taking the form of a slave, being born in human likeness. And being found in human form". Under the single heading "Form, Substance", NIDNT (Brown 1975, vol. I, pp. 703–10) "examines three words which may be translated as form:

*eidos*, *morphē* and *schēma*" (p. 703), highlighting their common semantic value of "visible appearance". Thus, "morphé is instanced from Homer onwards and means form in the sense of *outward appearance*" (*m.e.*, p. 705). This meaning is also literally attributed to *schēma*, presented as "*outward appearance*, form, shape" (*m.e.*). In this synonymization, the more robust 'empirical' connotation of *schēma* with respect to *morphē* prudently goes in the background, and the peculiarity of the term is absorbed by the *caveat* urging not to interpret it literally, because its correct understanding must be historicized: "In studying the Gk. word, one has to beware of the modern outlook which would relate *schēma* merely to external things, implying that the essential character was something different. To the Gk. mind, the observer saw not only the outer shell but the whole form with it." (p. 709). The hermeneutical unease inherent in this reductive equation is confirmed by the problematic choice of condensing the translations of ἐν ὁμοιώματι and σχήματι into the same word: "being born in the likeness [*schēmati*] of men." (ibid.) and its illustration: "This does not refer to the moral character of this earthly life (Lohmeyer), or to the appearance of Jesus (Dibelius), or to the fact of his humanity, but to the way in which Jesus' humanity appeared (Käsemann), as anyone could see. This is the force of *heuretheis* («being found»)." (ibid.) The distinction between "the fact of Jesus' humanity" and "the way in which this humanity appeared" is uniquely obscure. It complicates, rather than clarifies, the understanding of the Pauline text.

More differentiated are the respective entries in TDNT (1971, vol. VII, pp. 954–58) and BAGD, p. 872, which introduce an autonomous entry for *schēma* and do not erase its semantic autonomy but maintain its conceptual dependence on the notion of *morphē*. BAGD enunciates the double meaning of the following: "1 the generally recognized state or form in which someth. appears, outward appearance, form, shape of pers." and "2 the functional aspect of someth., way of life, of things" (ibid., p. 872) but without thematically problematizing the connotation of *exteriority*. TDNT points out the 'sensory' dimension: "σχήμα always denotes the outward form or structure perceptible to the senses and never the inward principle of order accessible only to thought. [...] it always ref. to what may be known from without" (p. 954), but it leaves its adoption in the text equally unexplained and binds itself in a hermeneutical impasse to which I will return later.

[7] For a comprehensive survey of the occurrences of the term *schēma* in classical literature and the Bible, in addition to the already mentioned BAGD, NIDNT, TDNT, cf. the corresponding entries in the LSJ (1843) and in the ancient but always valuable (Grimm and Wilke 1889; Trench [1858] 1880; Vincent 1887). The classic Auerbach ([1944] 1984) is also indispensable. For an interpretative overview, see the review in (Reumann 2008, p. 351): "in appearance. *schēma*, esp. since Lft. 127–33, changeable outward shape, contrasted with *morphē* (6th sphere) inner essential form. Michaelis 38, appearance versus essence. Further study suggested a less sharp contrast and a wider range of meanings for both terms. G. Harder, NIDNT 1:703–14 treats both under «form»; Braumann 709, avoid such distinctions between outer (shell) and inner (essential character). W. Pöhlmann, EDNT 3:318 lists other contrasts to be avoided (e.g., Loh. 1928, nature and «history»)".

[8] In the philosophical field, the word is articulated in two basic semantic cores. On the one hand (see, for example, the occurrence in Plato: "limit of solid" *Menon* 75a–76a (https://www.perseus.tufts.edu/hopper/text?doc=Perseus:text:1999.01.0178:text=Meno:section=76a, accessed 22 March 2024), it has the sense of *figure* as the designation of "sensible form" ("form that defines a body", as Dante sums up with his usual verbal felicity, cf. *Convivio* III IX 6). On the other hand, it has the sense of "structure": a form, a logical or more generally rational architecture, which is prominent, for example, in the Aristotelian terminology of the "figure of the syllogism" and the "structure of the comedy". Between these two semantic domains lies the meaning of *schēma* in the classical,

Greek and Latin, rhetorical tradition, which generally refers to the deviant processes of stylization and transgression of common speech described in the *elocutio*, the art of *schēmata* (figures) and *tropes* (stylistic manipulation and alteration of the semantic values of words), cf. Auerbach ([1944] 1984, pp. 11–28) and Genette (1966). The semantic value of the term in the rhetorical area may seem completely alien to Pauline usage (which is generally unambiguously traced back to the meaning of figure as sensible form), but in fact, it may help to reconstruct it more adequately. In its rhetorical sense, *schēma* generally designates the deviant processes of stylization and transgression of ordinary discourse, intercepting a peculiar "mobility" of meaning that can deviate from conventional lexical meanings by producing new semantic constellations. This connotation of mobility does not appear directly in philosophical usage, which tends to identify a structural (*schēmatic*) or sensible (figural) unity, but it is fundamental to the Pauline appropriation of the term.

Indeed, the hypothesis proposed here is that in 1 Cor 7:31; Philippians (2:7d and 3:21) and Rom 12:2, the philosophically pre-eminent connotation of the form of a sensible entity converges with the rhetorically marked connotation of mobility, coming to designate the exquisitely temporal dimension (passage, becoming) of the sensible form: its constitutive mutability.

9    Crucial here is the conceptual framework established by two seminal works such as (Trench [1858] 1880) and (Lightfoot 1878). For (Trench [1858] 1880, sct. LXX, pp. 261 ff.), the incarnation of the Son of God is in the *morphē* of a man, and *skhēma* simply denotes "the outward facts which came under the knowledge of his fellow-men", those who had personal contact with Jesus (hence *skhēma* has a "*superficial* character"). "The μορφή then, it may be assumed, is of the essence of a thing. We cannot conceive the thing as apart from this its formality, to use «formality» in the old logical sense; the σχῆμα is its accident, having to do, not with the «quidditas», but the «qualitas,» and, whatever changes it may undergo, leaving the «quidditas» untouched, the thing itself essentially, or formally, the same as it was before". The same synonymic assimilation is made by Lightfoot (1878, pp. 127ff.), who devotes a short chapter to "The synonimes μορφὴ and σχῆμα". On the one hand, Lightfoot captures the pivotal dimension of the term, which identifies the sensible dimension not as exteriority but as mutability, pointing out that in the *New Testament*, "This word retains the notion of instability, «changeableness» quite as strongly as in classical Greek" (ibid., p. 130), but on the other hand, this temporal nature is 'sterilized' by him in an interpretation closed within the dualistic–substantialist horizon of Greek metaphysics: "Thus in the passage under consideration the μορφή is contrasted with the σχῆμα, as that which is intrinsic and essential with that which is accidental and outward. And the three clauses imply respectively the true divine nature of our Lord (μορφῇ Θεοῦ), the true human nature (μορφὴν δούλου), and the externals of the human nature (σχήματι ὡς ἄνθρωπος)." (ibid., p. 133). From a cultural–historical point of view, Lightfoot's reading, like all those that follow it, is based on the hypothesis of a high degree of 'philosophical Hellenization' of the Hebrew Paul that is a more speculative than historically proven hermeneutical proposal: "we need not assume that St Paul consciously derived his use of the term from any philosophical nomenclature. Yet [...] the speculations of Alexandrian and Gnostic Judaism formed a ready channel, by which the philosophical terms of ancient Greece were brought within reach of the Apostles of Christ." (ibid.).

10   The linguistic choice of two biblical translations fundamental to modernity is exemplary: the *schēma* of 2,7d is translated by the gestural–corporeal "Gebärden", in contrast to the abstract "Gestalt", by Luther (LB) ("und nahm Knechtsgestalt an, ward gleich wie ein andrer Mensch und an Gebärden als ein Mensch erfunden": "and took upon himthe form of a servant, and was made just like any other man, and was found in his bodily expression as a man", *m.tr.*), while in the *King James Bible* (KJB), it is rendered as "fashion" vs. "form" ("took upon him the form of a servant, and was made in the likeness of men: And being found in fashion as a man") (a translation that is taken up in Martin 1997, p. 163).

Lightfoot (1878, pp. 112–13) strongly emphasizes the connotation of "external semblance" (erasing the connotation of mutability, which he had correctly pointed out): "In the present the opposition is between what He *is* in Himself, and what He *appeared* in the eyes of men; hence", the terms σχήμα, ὁμοιώματι, and εὑρεθεὶς are thus "all expressions implying external semblance". In line with this interpretation, see NABRE: "found human in appearance"; (Penna 2002, p. 43): "diventando partecipe degli uomini, / e, trovato all'apparenza come uomo" ("becoming a partaker of men, and, found in appearance as a human being", *m.tr.*); (Reumann 2008, p. 333): "born in humanity's likeness, and, in appearance perceived as a human being"; (Ehrman 2014): "And coming in the likeness of humans./ And being found in appearance as a human".

11   (BJ): "Devenant semblable aux hommes et reconnu à son aspect comme un homme" ("Becoming similar to men and recognized as a man by his appearance", *m.tr.*); (Fabris 2000, p. 104): "trovato nell'aspetto come uomo" ("found in the aspect as a man", *m.tr.*).

12   "Erscheinungsweise" is the translation suggested by Käsemann (1950, p. 339). In analogy to NDNT, Dunn (1998, p. 76) problematically translates *schēma* as a synonym for *homoíōma*: "and became in the very likeness of humankind. And being found in likeness as a human being". O'Brien (1991, p. 211) cuts the exegetical Gordian knot with a radical simplification that eliminates the term from the translation: "and being born like other human beings. And being recognised as a man".

13   Whether explicit or implicit, this bias is always at work, even in "functional" exegesis, which challenges the pericope's claim to elaborate a doctrinal Christology (for this reading, see Cullmann 1963).

14   On this point, cf. (Martin 1997, p. 203ff.). It is difficult to share the position of Marchal ([2014] 2017, pp. 17–18), for whom the need to sweep away possible docetic resonances of the pericope is an anachronistic preoccupation of contemporary exegetes.

15   Cf. Martin (1997, pp. 207ff.), followed by O'Brien (1991, pp. 231 ff.), who in translating σχήματι εὑρεθεὶς ὡς ἄνθρωπος as "appearance of a man" defends the view that this expression not only does not weaken but, on the contrary, strengthens the affirmation of Jesus' full human identity: "It states, without equivocation, the reality of His humanity" (ibid., p. 207).

16   In the Note to the verse (p. 2796), the BG defends this twofold linguistic choice in detail: "v 11.—*becoming like men*: there is no intention to attenuate the humanity of Jesus (Gal 4:4; Rom 1:3; 9:5; Heb 2:17). But if he was not different, he could not save us. He who was "living" (2 Cor 4:10–11) raised up those who were "dead" (Rom 6:4; Col 2:13). He did not need to be reconciled to God (2 Cor 8:9), whereas all others did (2 Cor 5:1819).—*By his recognized appearance as a man*: even though his way of being is different, Christ shares the human nature common to all men." (m.tr.). However, the argument is not without ambiguities: the equivalence of the predicate "to be *like men*" and "*to be a man*" is not at all obvious and the "nature" of an alleged diversity, which cannot be "of nature" (if Jesus, as recognized in the Nicene symbol, "shares human nature, common to all men") but is presented as functionally necessary ("if he was not different, he could not save us"), remains entirely indeterminate and equivocal. If the postulation of an ontological diversity of the man Jesus is anti-Nicene (Jesus is true God and true man), the postulation of a purely moral diversity is not Christologically permissible, because the man Jesus "did not need to be reconciled with God". So, the question remains: does this double messianic power (of life and of communion with God) make him different as a man (*different from men*, only *similar to them.* In which case it is a difference of "nature")? Or does it make him different just as an individual (in this case, it is an eschatologically constitutive historical difference) who, through his own death and resurrection, initiates the possible transfiguration of all human beings into a new form of humanity (raising them to be "the sons of God", Phil 3:21)? The assertion that "his way of being is different" is then simply a tautological repetition of the statement, leaving the fundamental question of its content unanswered.

   While avoiding the trap of translating the noun "likeness" with the adjective "similar" ("coming in human likeness"), NABRE problematically opts (as already noted) for the insidious (docetic) lexical choice of "appearance": "and found human in appearance". The New English Version of the *Jerusalem Bible* (RNJB) and ESV avoid these problems by adopting a non-literal and anodyne synonym translation of *schēma* to *morphē* (respectively: "born in human likeness and found in human shape"; "being born in the likeness of men. And being found in human form"). Similarly, the *Einheitsübersetzung* (EU), the standard version of the German-speaking Catholic Church, opts for a free translation: "[wurde] den Menschen gleich./Sein Leben war das eines Menschen" ("[he became] equal to men / His life was that of a man", m.tr.), relegating the literal one to the footnote ("wurde den Menschen gleich / und der Erscheinung nach ganz als Mensch erfunden").

17   See the hammering iteration of *metaschēmatizei* with the meaning of *masquerade* in 2 Corinthians 11: 13–15 ("For such people are false apostles, deceitful workers, who masquerade as apostles of Christ. And no wonder, for even Satan masquerades as an angel of light. So, it is not strange that his ministers also masquerade as ministers of righteousness"). In this triple occurrence, the semantic connotation activated is that of the sensible form as a visible manifestation, as an exteriority, which allows for the fraudulent exploitation of appearance as an illusionistic effect. This negative connotation of the sensible form as exteriority, to which Paul also resorts, is in direct contrast to the positive connotation that the semanteme assumes in Phil 3:21 and Rom 12:2, where the sensible form does not denote outward appearance but a mutability that is eschatologically sublimated into spiritual (Rom 12:2) and bodily transfiguration (Phil 3:21).

18   Or the author/s of the pericope attributed to him, included in that collection of Pauline texts that some exegetes believe to be the Epistle to the Philippians. On the question of the discontinuous and composite structure of the letter, which may therefore be read as an assemblage of textual fragments (possibly three epistolary bodies), (cf. O'Brien 1991, pp. 47ff., 206ff.; Reumann 2008).

19   "For the world in its present form is passing away" (NABRE).

20   "The whole classical tradition was very much alive in St. Augustine, and of this his use of the word *figura* is one more indication. In his writings we find it expressing the general notion of form in all its traditional variants, static and dynamic, outline and body; it is applied to the world, to nature as a whole, and to the particular object; along with *forma, color,* and so on, it stands for the outward appearance (*Epist.*, 120, 10, or 146, 3); or it may signify the variable aspect over against the imperishable essence. It is in this last sense that he interprets I Cor. 7:31: *Peracto quippe iudicio tune esse desinet hoc coelum et haec terra, quando incipiet esse coelum novum et terra nova. Mutatione namque rerum non omni modo interitu transibit hic mundus. Unde et apostolus dicit: praeterit enim figura huius mundi, volo vos sine sollicitudine esse. Figura enim praeterit, non natura* (*De civitate Dei*, 20, 14). ("When the judgment shall be finished, then this heaven and this earth shall cease to be, and a new heaven and a new earth shall begin. But this world will not be utterly consumed; it will only undergo a change; and therefore the Apostle says: The fashion [figura] of this world passeth away, and I would have you to be without care. The fashion [*figura*] goes away, not the nature.") [Trans. John Healey, Everyman edition. London, 1950, Vol. II, p. 289.]" (Auerbach [1944] 1984, p. 37).

21   In so far as (1) creation is not instantaneous but gradual: the ontological architecture of the cosmos is transformed in the different days—stages—of the creative action of God; (2) creation is not God's last word on cosmos: this is said with the resurrection of his Son, as the condition of the universal and definitive establishment of his sovereignty.

22   On the need to think of the *cosmos* as an "*eschatological* concept", more temporal than spatial, material, see (Bultmann 1951): "Now this means that "cosmos"—used in the above sense—is much more a time-concept than a space-concept; Nor, more exactly, it is an *eschatological concept*. It denotes the world of men and the sphere of human activity as being, on the one hand a temporary thing hastening towards its end (I Cor. 7:31), and on the other end, the sphere of anti-godly power under whose sway the individual who is surrounded by it has fallen." (ibid., p. 256). While Bultmann emphasizes the temporal (and not substantivist–metaphysical, as in the Greek philosophical tradition) nature of the Pauline notion of the *cosmos*, pointing out that Paul uses the expression "this world" as a synonym for "this time", he nevertheless does not resist reinstating the term *schēma*, carefully chosen by Paul to avoid this metaphysical identification, as "essence": "The present is characterized by the sentence: «the *schema* (essence) of this

world is passing away»" (I Cor.7: 3b). /.../ «This world» can also interchange with «this age» (αἰών). /.../ «The *schema* of this world» (I Cor. 7: 31) is «the present evil age » of Gal. 1: 4». (ibid.). For Paul, not all the present is *schēmatic* but only the present of the pre-messianic time, which is subject to sin. The notion of sin is linked by Paul (and by John) to that of the world as the historical and eschatological condition of man, because it implies an "understanding of man's situation as an enslavement to power for whose dominion he nevertheless is himself responsible" (ibid., p. 257).

23   "For the law of the Spirit, which gives life in Christ Jesus, has delivered you from the law of sin and death" (Rom 8:2).

24   Temporality is not the direct cause of the corruption of entities but its agent (cf. Aristotle 1930, Physics IV 12–13): "A thing, then, will be affected by time, just as we are accustomed to say that time wastes things away, and that all things grow old through time, and that there is oblivion owing to the lapse of time, but we do not say the same of getting to know or of becoming young or fair. For time is by its nature the cause rather of decay, since it is the number of change, and change removes what is." (IV 12, 221a-b) "In time all things come into being and pass away [...]. And this is what, as a rule, we chiefly mean by a thing's being destroyed by time. Still, time does not work even this change; even this sort of change takes place incidentally in time". (222b).

25   NABRE has "futility", but I follow (KJB) and other translations which choose the traditional term "vanity".

26   The argument presented here takes up the Adamic analogy for the Philippian Jesus evoked by Dunn ([1980] 1989, 1998) in terms of "enslavement to corruption and sin" and "submission to death". However, it does not embrace the interpretation of the synonymous use of *morphē* and *eikōn* nor the opposition between the man Jesus and the man Adam in terms of the temptation (victoriously resisted by Jesus) to violently "appropriate" equality with God, in a key that reads the claim of pre-existence as uniquely human. (For a discussion of the Adamic parallelism thesis and the question of pre-existence, see O'Brien 1991, pp. 264ff.; Martin 1997, pp. 99ff.). Cf. (Marchal [2014] 2017, pp. 20–21), for a radically alternative reading to the one proposed here. For this author, the pericope (Philippians in general) has nothing to do with the power of sin and death: "Sin, in fact, is a major preoccupation of this section of Romans [5:12–21], whereas it is not even a minor topic in this or any part of Philippians!".

27   Liberation from "this body of death", the flesh (σὰρξ), which enslaves to the "law of sin" (Rom 7:23–25), in the establishment of the sovereignty of the "law of the Spirit" (Rom 8:2), which makes "slaves of God" (Rom 6:20–23).

28   On the corresponding entry, (Spicq 1994) notes that it is "wrong" to translate *doulos* as "servant", because it is a technical designation not of a service function but of a social status of "proprietary" subordination.

29   In Rom 12:2, the temporal connotation of the semanteme is clearly activated, with the indication that the spiritual *metamorphosis* (μεταμορφοῦσθε) produced by the conversion is realized as de-*schēmatization, de-figuration* (μὴ ʳσυσχηματίζεσθε): detachment from the temporal form of the present world (τῷ αἰῶνι τούτῳ: *huic seculo*): "*Do not conform yourselves to this age but be transformed by the renewal of your mind, that you may discern what is the will of God, what is good and pleasing and perfect*" (*m.e.*).

30   It is impossible to recapitulate here the endless exegetical discussion on the reasons for the possible alternative translation and interpretation choices for each individual term in the pericope. Since the specific objective of this paper is to determine the meaning of the word σχήμα, only the translation issues directly related to this point will be briefly analyzed here, and the NABRE translation will be adopted, modified only in the locutions of verses 6a-b and 7c-d, which are highlighted in bold. In my opinion, the Latin translation of the VC, which I quote for comparison, remains illuminating for a correct interpretation of the pericope:

"Hoc sentite in vobis, quod et in Christo Iesu:

qui cum in forma Dei esset, non rapinam arbitratus est esse se aequalem Deo,

sed semetipsum exinanivit formam servi accipiens,

in similitudinem hominum factus;

et habitu inventus ut homo,

humiliavit semetipsum factus oboediens usque ad mortem,

mortem autem crucis."

31   ὃς ἐν μορφῇ Θεοῦ ὑπάρχων—without discussing the merits of the alternative interpretations (cf. Reumann 2008, pp. 333ff., for an analytical review of all the thorny exegetical, theological, and lexical issues inherent in these verses), here I adopt Vincent's old choice: "Being in the form of God (ἐν μορφῇ Θεοῦ ὑπάρχων)
Being. Not the simple εἶναι to be, but stronger, denoting being which is from the beginning. See on James 2:15. It has a backward look into an antecedent condition, which has been protracted into the present. Here appropriate to the preincarnate being of Christ, to which the sentence refers. In itself it does not imply eternal, but only prior existence."
(Vincent 1887, http://biblehub.com/commentaries/philippians/2-6.htm, accessed 22 March 2024).
The question of whether this locution can be read as a Pauline assertion of either a consubstantial or similar divine pre-existence of Jesus is the subject of an open and extremely sensitive debate, theologically highly significant (for the thesis of divine similar pre-existence, with different nuances, see Dunn [1980] 1989, 1998; Martin 1997).

32   The exegetical and theological–philological reconstruction of the term μορφὴ occupies a prominent place in the studies of the pericope (cf. the critical–bibliographical survey in O'Brien 1991, pp. 215ff.; Martin 1997, pp. 99ff.; and Hawthorne 1998).
From my point of view, it is essential to avoid the parasynonymic assimilation of σχήμα and μορφὴ, which obscures the crucial point that *doulos* is not referring to the human being but to the creature in general. As I have tried to show in the previous section,

"taking the form of a slave" means entering "the sphere" (cf. Reumann 2008) of the cosmos, assuming the status of a creature, by incarnating as a human being: being born (γενόμενος) in the pre-messianic bodily form (*schēma*) of a human being. In other words, "the form of a slave" does not yet denote the assumption of the *human form* (the birth into the *schēma*, the carnal body of man, expressed in 2, 7c-d) but its precondition in the emptying of the Creator into the creature.

In summarizing the reasons for the translation choice of *morphē* as "sphere", Reumann (2008, p. 344) quotes a summary by H.J. Kuschel, which is also useful in illuminating some of the qualifying points of this reflection: "As Kuschel 606 n 46 put it, "Anyone who decides . . . for «appearance» . . . runs the risk of reading into the text a contrast between changing «external appearance» and a permanent «inner being.». . . Anyone arguing that this is a statement about Christ's nature» runs the danger that such a statement about Jesus «can be misunderstood in physical-real terms». Anyone for status, position (Schweizer) «will hardly find a parallel in other New Testament writings" (Gnilka 113–14). Anyone for «divine glory» (Schnackenburg 1970, p. 315) overlooks the fact that in the hymn «the obedient one only received this status after the humbling and not before». Kuschel, Käsemann, and Translation opt for sphere (realm, place and relationships)". Avoiding the Gnostic implications associated with his interpretation, Käsemann's (1950, p. 321) concluding formulation is, in my view, convincing: "Unter μορφή ὕϵον bzw. δούλου ist dann einfach die himmlische bzw. irdische Daseinsweise zu verstehen".

[33] As postulated, instead, by the majority of exegetes, who also give different interpretations to this identification of the slave as the form of the human being, in an arc of readings that are distributed between the Adamic parallel, the biblical figure of the Just Sufferer, and the evocation of the Servant of Isaiah (for a bibliographical overview of the different readings, cf. O'Brien 1991, pp. 224ff.; Martin 1997, pp. 169ff.; Reumann 2008, pp. 335ff.; Marchal [2014] 2017, pp. 21ff. for new exegetical proposals).

In partial agreement with the hypothesis put forward here, Käsemann's (1950) interpretation identifies bondage precisely as a condition of cosmic subjugation, but on the one hand, he charges it with a mythical, mystery component, of a Gnostic and Hellenistic matrix, which does not belong to the Pauline text. On the other hand, he insists on the identification of the *doulos* as a human being and not as a creature in general. By emphasizing (in direct controversy with Lohmeyer 1961) the soteriological–kerigmatic (not purely ethical) dimension of the pericope, within the cosmic horizon of the exaltation of Christ as "Pantokrator", Käsemann overshadows the redemptive significance of the exaltation of the Son elevation, which takes place precisely through his resurrection. The "Knechtschaft durch die Mächte" ("Bondage by the powers") referred to by Käsemann (1950, p. 345) plays a distinctly secondary role for Paul, because for him, the absolute power that enslaves creation is physical and moral death, the epitome of transience, insofar as it is the vector of corruption that is consummated as sin and erosion, the cessation of life. It is "the law of sin and death" (Rom 8:2) that enslaves the human beings and from which Christ sets them free (Rom 6:20–23). Only in this framework can we understand why the kenotic humiliation of the One who, being equal with *God*, made himself equal with the creature, exposing himself to death, is the condition of his exaltation. The Christ, as the Risen One victorious over death, in his capacity is recognized by all creatures as the "Pantokrator", who establishes a new lordship, one of freedom and not of bondage ("I no longer call you slaves [...] I have called you friends", Jn 15:15).

[34] As noted above, it should not be forgotten that when Paul speaks of *kosmos*, he is using a Greek term to give it a meaning which is specific to the Jewish–biblical tradition. On the difference between the Greek concept of the *kosmos* as a totality unified by rational laws, encompassing heaven, earth, and all living beings, including humans and God, and the Testament concept, see again Bultmann (1951, pp. 254ff.). The *Old Testament* "does occasionally speak of the «all» and, much oftener, of «heaven and Earth»—but always in such a way that God himself is not included in it, but is always distinguished from it as the Creator. In this restricted sense, Hellenistic Judaism took over and used the term «cosmos», and it is in this sense that the New Testament, inclusive of Paul, uses it. /…/ However, «kosmos» does not always mean «earth» as the mere stage for man's life and living but oftem denotes the quintessence of earthly conditions of life and earthly possibilities. It embraces all the vicissitudes included between the pairs of polar terms «life. . .death», «things present . . .things future» (I Cor. 3: 22). Accordingly, human life in its worldly aspects, in its hustle and bustle, in its weal and woe, is a «dealing with the world» (I Cor. 7: 31)—and as the antithesis to the «affairs of the world», the «affairs of the Lord» hover in the background (7: 32–34; see §22)" (ibid., p. 254). The opposition between the *kosmos* and its Creator (as a division between the dominion condition of sin and the holiness of God) is central for John, who repeatedly proposes it; cf. the passage in 1 Jn 2: 15-18, which is a kind of paraphrase of 1 Cor 7: 29–31, and which invites not to " love the world or the things of the world", [...] for all that is in the [...] is not from the Father but is from the world. Yet the world and its enticement are passing away".

[35] ὃς ἐν ταῖς ἡμέραις τῆς σαρκὸς: *in diebus carnis suae.* (Heb 5: 7)

[36] In the light of these passages from *Philippians* and 1 *Corinthians*, the extent of the debt that the existential analysis of Heidegger's *Being and Time* owes to the Pauline theology is evident. To "be found" (*inventus*, in the *Vulgata*) in the temporal figure (*skhēma: habitus*) of man is, in Heideggerian terms, *Being-in-the-World*, to dwell in the world (Heidegger [1927] 1977, sct. 12), as *Dasein* (the being to which the world opens as *Attunement*, §29), the being whose essence consists in its existence (§9). To be *skhēma hōs anthrōpos* is to be part (transcendent in the permanent anticipation of death) of the *skhēma tou kosmou*: the essential constitution of *Dasein* is *Being-in-the-world* (§28) in the dismissal ("falling and throwness", §38) of facticity, finding itself exposed to contingency, subject to the condition (*cosmic, worldly*: intrinsic to temporal finitude) of *Being-toward-Death* (§§46-5). *Dasein* as *Being-in-the-World* transcends itself, *surpasses itself* (*Being-ahead-of-itself*) in its openness to annihilation as the most authentic condition of being.

[37] The *Septuagint* translates the original Hebrew term using the same semanteme as Paul: "Then God said: «Let us make human beings in our image, after our likeness (ὁμοίωσιν)»". The Vulgate lexically emphasizes the parallelism by translating the two

different Greek verbs of Gen 1:26 (Ποιήσωμεν) and Phil 2:7c (γενόμενος) with the same term, "*facere*": "ait faciamus, hominem ad imaginem et similitudinem nostram"; "in similitudinem hominum factus". The Vulgate model is followed by Martin (1997, p. 163), who translates the following: "and *was made* in the likeness of men" (*m.e.*). Reumann (2008, p. 349) recalls that "-ma [stands] for the result of an action". (For a philological–exegetical discussion of the term, see Martin 1997, pp. 199ff.).

38    This concept is reiterated literally in Heb 4:15: Jesus is a high priest who sympathizes with our weaknesses, because he was "one who has *similarly* (based on his likeness with us) been tested in every way, yet without sin ("temptatum autem per omnia *pro similitudine* absque peccato": "πεπειρασμένον δὲ κατὰ πάντα καθ᾽ ὁμοιότητα χωρὶς ἁμαρτίας", *m.e.*).

39    It is important to stress that the semanteme of *homoiōsis* and *homoíōma* is different from *mímēsis*: it expresses a similarity, an analogy, which is not established by imitation but by participatory assimilation. It does not denote iconic parallelism but processual proximity. In this interpretive key, the Adamic reference does not pass through the equivalence of *morphē* with *eikōn* (image) but through the notion of *likeness* (κατ᾽ εἰκόνα ἡμετέραν καὶ καθ᾽ ὁμοίωσιν) and is not specified in a human dramaturgy of temptation (which Jesus, in contrast to Adam, resisted, cf. Dunn [1980] 1989, 1998) but in a soteriological view of God's self-giving in the incarnation.

40    The pericope expresses with eloquent clarity and solemn jubilation the universal, cosmological, and not merely human dimension of the Redemption: all creatures bow before the sovereignty of the One who has freed them—"every knee should bend".

41    "I mean that as long as the heir is not of age, he is no different from a slave, although he is the owner of everything, but he is under the supervision of guardians and administrators until the date set by his father. In the same way we also, when we were not of age, were enslaved to the elemental powers of the world (ὑπὸ τὰ στοιχεῖα τοῦ κόσμου ⌐ἤμεθα δεδουλωμένοι. But when the fullness of time had come, God sent his Son, born of a woman, born under the law, to ransom those under the law, so that we might receive adoption. As proof that you are children, God sent the spirit of his Son into our hearts, crying out, "Abba, Father!" So you are no longer a slave but a child (ὥστε οὐκέτι εἶ δοῦλος ἀλλὰ υἱός), and if a child then also an heir, through God." (Gal 4:1–7).

42    The "world to come" (*Olam Ha-Ba*) is a key concept in Hebrew eschatology to denote the coming of the Messiah (a central proclamation in Isaiah, cf. especially: Is 2:11, 25, 51–53) and the condition of the resurrected (cf. Is 26:19 Hos 6:1–3; Ezek 37:1–14; Job 14:13–15). In the *New Testament*, the contrast "between this age and the age to come" is formulated as a "temporal" difference, retained in the Latin translation but often deleted in translations that spatialize the original Greek *aiōn* into *world* and thus "disfigure" the temporal condition referred to. Cf. τῷ αἰῶνι οὔτε ἐν τῷ μέλλοντι: *in hoc saeculo neque in futuro* of Mt 12:32 ("either in this age or in the age to come"). See also Lk 20:35 and Eph 1:21.

## References

### Quoted Bible Translations

*Bibbia di Gerusalemme.* 2011. Bologna: Edizioni Dehoniane.

*Bible de Jérusalem.* 2003. Paris: Cerf

(CEI 2008) Conferenza Episcopale Italiana. 2008. La Sacra Bibbia (https://www.lachiesa.it/bibbia/, accessed 22 March 2024).

*Einheitsübersetzung.* 2016. Katholisches Bibelwerk (https://www.bibleserver.com/EU/Philipper2, accessed 22 March 2024).

*King James Bible* (https://www.biblegateway.com/passage/?search=Philippians%202&version=NABRE, accessed 22 March 2024).

*Lutherbibel* (https://bibeltext.com/l12/philippians/2.htm, accessed 22 March 2024).

*New American Bible Revised Edition.* 2010 (bible.usccb.org/bible, accessed 22 March 2024).

*Novum Testamentum Graece.* 2012. Nestle-Aland Greek New Testament. 28th Revised Ed. with Critical Apparatus. Deutsche Bibelgesellschaft. Peabody, Massachusetts: Hendrickson Publishers.

*The Holy Bible English Standard Version. Catholic Edition.* 2007 (https://www.biblegateway.com/passage/?search=Philippians+2&version=ESV, accessed 22 March 2024).

*The Revised New Jerusalem Bible.* Edited by Henry Wansbrough. New York: Image 2019.

*Vulgata Clementina* (https://bibeltext.com/vul/philippians/2.htm, accessed 22 March 2024).

### Secondary Literature

Aristotle. 1930. *Physica. The Works of Aristotle*. Translated under the editorship of William David Ross. Oxford: Clarendon Press, vol. II.

Auerbach, Erich. 1984. Figura. In *Id. Scenes from the Drama of European Literature*. Translated by Ralph Manheim. Minneapolis: University of Minnesota Press, pp. 11–76. First published 1944.

Bird, Michael F., and Nijay K. Gupta. 2020. *Philippians*. Cambridge and New York: Cambridge University Press.

Brown, Colin, ed. 1975. *The New International Dictionary of New Testament Theology*. Grand Rapids: Zondervan.

Bultmann, Rudolf. 1951. *Theology of the New Testament*, 3rd ed. New York: Charles Scribner's Sons, vol. I.

Cullmann, Oscar. 1963. *Die Christologie des Neuen Testaments*. London: SCM Press.

Dunn, James D. G. 1989. *Christology in the Making*, 2nd ed. London: SCM Press. First published 1980.

Dunn, James D. G. 1998. Christ, Adam, and Preexistence. In *Where Christology Began: Essays on Philippians*. Edited by Ralph P. Martin and Brian Dodd. Louisville: Westminster John Knox Press, pp. 74–83.

Ehrman, Bart D. 2014. *How Jesus Became God: The Exaltation of a Jewish Preacher from Galilee*. New York: HarperOne.

Fabris, Rinaldo. 2000. *Lettera ai Filippesi. Lettera a Filemone*. Bologna: EDB.

Fewster, Gregory P. 2015. The Philippians 'Christ Hymn': Trends in Critical Scholarship. *Currents in Biblical Research* 13: 191–206. [CrossRef]

Fitzmeyer, Joseph A. 1988. The Aramaic Background of Philippians 2:6–11. *The Catholic Biblical Quarterly* 50: 470–83.

Genette, Gérard. 1966. *Figures I*. Paris: Seuil.

Grimm, Carl Ludwig Wilibald, and Christian Gottlob Wilke. 1889. *Greek-English Lexicon of the New Testament*. Translated, revised, and enlarged by Joseph Henry Thayer. New York: American Book Company.

Hawthorne, Gerald F. 1998. *The Form of God and Equal with God (Philippians 2:6)*. Edited by Ralph P. Martin and Brian J. Dodd. Louisville: Westminster John Knox Press, pp. 96–110.

Heidegger, Martin. 1977. *Sein und Zeit*. Hrsg. von Friedrich-Wilhelm von Herrmann. Gesamtausgabe, Band 2. Frankfurt am Main: Vittorio Klostermann. First published 1927.

Käsemann, Ernst. 1950. Kritische Analyse von Phil. 2,5-11. *Zeitschrift für Theologie und Kirche* 47: 313–60.

Lightfoot, Joseph Barber D.D. 1878. *Saint Paul's Epistle to the Philippians*. A Revised Text with Introduction, Notes, and Dissertations. Cambridge: MacMillan.

Lohmeyer, Ernst. 1961. *Kyrios Jesus: Eine Untersuchung zu Phil. 2, 5-11*, 2nd ed. Heidelberg: Carl Winter, Universitätsverlag.

LSJ. 1843. *Middle Liddell. Greek—English Lexicon*. Edited by Henry George Liddell, Robert Scott, Henry Stuart Jones and Roderick McKenzie. Oxford: Oxford University Press. Available online: https://www.perseus.tufts.edu/hopper/morph?l=sxh/mati&la=greek&can=sxh/mati0&prior=kai%5C&d=Perseus:text:1999.01.0155:book=Philippians:chapter=2:verse=7&i=1#lexicon (accessed on 22 March 2024).

Marchal, Joseph A. 2017. *Philippians: An Introduction and Study Guide*. London and New York: Bloomsbury. First published 2014.

Martin, Ralph P. 1997. *A Hymn of Christ: Philippians 2:5–11 in Recent Research and Interpretation*. Downers Grove: InterVarsity Press.

Martin, Ralph P., and Brian J. Dodd, eds. 1998. *Where Christology Began: Essays on Philippians*. Louisville: Westminster John Knox Press.

O'Brien, Peter T. 1991. *The Epistle to the Philippians*. NIGTC. Grand Rapids: Eerdmans.

Penna, Romano. 2002. *Lettera ai Filippesi. Lettera a Filemone*. Roma: Città Nuova.

Reumann, John. 2008. *Philippians: A New Translation with Introduction and Commentary*. The Anchor Yale Bible. New Haven and London: Yale University Press.

Spicq, Ceslas. 1994. *Theological Lexicon of the New Testament*. Translated and Edited by James D. Ernest. Carol Stream: Tyndale House Publishers, vol. 1.

TDNT. 1971. *Theological Dictionary of the New Testament Theology*. Grand Rapids: Eerdmans, vol. VII (Σ).

Trench, Richard Chevenix D. D. 1880. *Synonyms of the New Testament*, 9th ed. London: MacMillan. First published 1858.

Vincent, Marvin. 1887. *Vincent's Word Studies in the New Testament*. New York: Charles Scribner and Sons.

