# Peer review of "Schēma: A Semantic Puzzle—Some Hermeneutical and Translational Difficulties about Philippians 2:7d"

_religions, doi:10.3390/rel15050613_

Round 1
Reviewer 1 Report
Comments and Suggestions for Authors
The argument has promise for shedding light on our understanding of schema and its theological ramifications. However, for the reasons cited below, I do not believe the article is ready for publication.
The citations bear witness to the considerable research behind his article. Even so, the footnotes include few citations of scholarship within the last ten years.
Comments on the Quality of English LanguageThis might be the primary problem that contributes to the obscurity of the thesis and argument. Convoluted sentence structure throughout the paper thwarts reading comprehension. Long strings of prepositional phrases and conditional clauses—separated by commas and parenthetical asides—bury the meaning of sentences in a thicket of over-qualification.
For example, lines 293–302 are one long sentence that comprise the whole paragraph. These lines could easily be divided into multiple sentences, and that restructuring would clarify the flow of thought and the argument being built.
The entire essay would need to be revised in line with the suggestions for lines 293–302 before I believe it would be ready for publication.
Author Response
I acknowledge the merits of the concerns raised in the review, and following the advice received, I have subjected the text to a robust linguistic reformulation based on the rereading done by an English-speaking scholar.
In the new version of the article that I am submitting, all the changes are marked in the track-change modality.
I hope that this intervention will contribute significantly to the clarification of the thesis and its argumentative coherence.
I note that the scarcity of "citations of scholarship within the last ten years" is due to the fact that recent research has not brought relevant novelties to the specific point addressed in the article, focusing on other aspects.
The many translation choices on this lexical item reported in the article show that the standard is still inertially set by the previous philological results summarized in the various Bible dictionaries and commentaries cited. One of the aims of this paper is precisely to question this continuity, just as it has been done in relation to other historical and theological issues that are at the center of today’s exegetical discussion.
Reviewer 2 Report
Comments and Suggestions for Authors
I have no suggestions. I think the article is valid and valuable as it stands. The proposed interpretation of a complex term offers a broader look at a Pauline text, opening up to a cosmic and eschatological perspective.
Author Response
I thank you for the sympathetic reading and positive evaluation of the proposed thesis.
Reviewer 3 Report
Comments and Suggestions for Authors
The premise, argument and playout of this paper was sensible and relatively straightforward as I followed it. As someone less familiar with Roman Catholic theological tradition and its relation e.g. to classical literature, I had to work hard sometimes at following the way it went about the task. It nevertheless made its case. My major confusion was around the word "lessical" in the title; I was expecting "lexical," since it centered on the translation and meaning of the one word in Grk, schema. Thus lessical will need to be clarified at the beginning of the paper, as to its meaning and purpose, as I do not know it from my language studies, linguistics, nor the any dictionaries I consulted. It seems to be getting at an aspect of language that was not evident to me. Thank you.
Comments on the Quality of English LanguageThe ms is clean and other than what I noted above with respect to the confusing (to me) word lessical, all else was in place and not in need of editing. The writing style is dense, and does assume a level of familiarity with some theological tradition with which I am not immediately working with.
Author Response
I thank you for your positive reading of the text, which I have since subjected to a thorough linguistic revision, correcting also the title according to your critical remark.
Round 2
Reviewer 1 Report
Comments and Suggestions for Authors
The thesis and the argument are much clearer after the revisions. The reading of schema addresses a translational difficulty and offers a novel and theologically reasoned solution.
After reading, I think the title of the paper is misleading. Calling it a "lexical note" leads the reader to expect lexical analysis. The paper's argument depends on theological commitments more than lexical analysis, and its primary contribution is theological rather than lexical. That is to say, the paper is more about the meaning of the incarnation than it is about this one word.
Comments on the Quality of English LanguageThe essay is considerably more readable after the revisions.
Author Response
I thank you for your renewed attention and hope that this rewording of the title will satisfy the objection raised:
Schēma: A Semantic Puzzle.
Some Hermeneutical and Translational Difficulties about Filippians 2:7d
